# Drivers and Barriers to Substituting Firewood with Biomass Briquettes in the Kenyan Tea Industry

Amalia Suryani [1,2,*] , Alberto Bezama [3,*] , Claudia Mair-Bauernfeind [4] , Macben Makenzi [5] and Daniela Thrän [1,3,6]

1   Institute for Infrastructure and Resource Management, Universität Leipzig, Grimmaische Str. 12, 04109 Leipzig, Germany; daniela.thraen@ufz.de
2   Faculty of Electrical Engineering, Mathematics and Computer Science, University of Twente, Drienerlolaan 5, 7522 NB Enschede, The Netherlands
3   Department of Bioenergy, Helmholtz-Centre for Environmental Research (UFZ), Permoserstraße 15, 04318 Leipzig, Germany
4   Institute of System Science, Innovation and Sustainability Research, University of Graz, Merangasse 18/I, 8010 Graz, Austria; claudia.mair@uni-graz.at
5   Deutsche Gesellschaft für Internationale Zusammenarbeit (GIZ) GmbH, Riverside Drive, Nairobi 00100, Kenya; macben.makenzi@giz.de
6   Deutsches Biomasseforschungszentrum Gemeinnützige GmbH, 04347 Leipzig, Germany
*   Correspondence: amalia.suryani@web.de (A.S.); alberto.bezama@ufz.de (A.B.)

**Abstract:** The tea industry in Kenya is among the main consumers of firewood for its intensive thermal energy demand. Along with the growing concerns about firewood depletion, tea factories have begun transitioning to alternative fuels to power their boilers. Briquettes made of biomass residues are among the promising solutions; however, they are not yet widely adopted. This study was conducted to identify the factors that motivate the tea factories to use biomass briquettes instead of firewood and the factors hindering such substitution. The substitution potential was assessed, and the drivers and barriers of the substitution were examined using a combination of SWOT (strengths, weaknesses, opportunities, and threats) analysis and a PESTEL (political, economic, social, technological, environmental, and legal) framework. The findings suggest that even though using biomass briquettes is technically possible, it is not economically favorable for tea factories. The SWOT/PESTEL analysis identified 27 factors influencing the substitution. Among the key drivers are the depleting supply of firewood, the availability of biomass residues, and the external support from development organizations to improve the technical capacity in both tea and briquette industries. The study revealed the barriers to substitution include the cost competitiveness, insufficient supply, and varying quality of briquettes, as well as the lack of awareness and knowledge of briquettes.

**Keywords:** biomass briquettes; firewood substitution; tea industry; drivers and barriers; SWOT analysis; PESTEL analysis; thematic analysis

## 1. Introduction

Sustainable management of forests and halting deforestation are among the targets of Sustainable Development Goal (SDG) 15. There is an alarming decline in the proportion of the forest area to the world's total land area from 31.9% in 2000 to 31.2% in 2020, where a significant decrease took place in Latin America and Sub-Saharan Africa [1]. In Kenya, the loss of primary forests reached 7.6% from 2002 to 2020, which was driven, among others, by the demand for cheap energy [2,3].

The share of bioenergy generated from traditional biomass, such as charcoal and firewood, has hardly changed over the last 25 years and accounts for approximately 60% of the total energy use in Kenya [4,5]. As the world's third-largest producer of black tea [2,6], the Kenyan tea industry requires around one million tons of firewood every year [7], mainly

to reduce the moisture content of the tea leaves [5,8,9]. The high demand for fuelwood by rural communities and industry has led to deforestation and the depletion of trees as a natural resource [3].

The Government of Kenya addressed this issue by imposing a forest harvesting moratorium firstly in 2018. Due to the moratorium, the supply of wood materials has been adversely affected, causing scarcity and increased market prices for wood products. Tea factories, which are firewood dependent, suffered additional expenditures in their production and, consequently, a decline in profits [10]. Therefore, they seek to invest in alternative energy sources.

Several studies have already investigated alternative energy sources for the tea sector [8,9,11,12]. Wind energy can be economically viable in several locations of tea factories in Kenya, although this solution addresses the electricity demand and does not offer firewood replacement [8]. Different solar drying technologies were investigated to dry different types of tea, which can improve its color and aroma ss [11]. However, the study did not provide insights on the feasibility of installing the technology in certain geographical settings [11]. A study in India [9] investigated different renewable energy technologies, i.e., solar, wind, hydropower, and bioenergy waste. The authors proposed a solar-biomass hybrid system for the drying and withering process [9]. Briquettes from rice husks and tea waste are also promising for tea industry application, although further studies are required to determine the optimum combustion properties [12]. There are also challenges to implementing these technologies in the tea sector. To name a few, the harvesting and transportation of biomass are energy-intensive, and storage space is needed; solar and wind energy is influenced by their diurnal and stochastic nature. Thus an appropriate site selection is important, and hydropower suffers from seasonal climatic changes that influence the water availability [9].

The tea factories in Kenya recognized the potential of biomass residues to substitute firewood in their processes. Kenya has a promising bioenergy opportunity from crop residues, for example, husks and bagasse, from the ongoing agricultural production such as maize, cassava, and sugarcane [5]. Briquettes have been introduced in several tea factories through cofiring, i.e., mixing firewood and briquettes in the boilers. However, this potential is not yet widely used. Recent reports suggest that operational problems, cost intensity, and the distance from briquette factories to tea factories can be some of the factors hindering the briquette use [7,10]. A review of the briquette sector in East Africa identified several factors impeding the advancement of the briquette use, namely the lack of specific regulations relating to briquettes, limited access to financing options, insufficient marketing and distribution strategies, and inconsistent supply of raw materials [13]. Furthermore, feedstock availability, lack of technical capacity, high cost of briquettes [14], lack of awareness, limited fiscal incentives, and lack of an overarching institutional framework [15] can be other challenges in implementation. However, to the best of our knowledge, no study has attempted to look specifically at the factors that influence the application of briquettes made of crop residues (hereinafter referred to as biomass briquettes) to be used as a substitution for firewood in the tea industry.

Against this background, this research aims to identify the drivers and barriers to substituting firewood with biomass briquettes in the Kenyan tea industry. We focus on the substitution potential and synthesizing the drivers and barriers within the PESTEL (political, economic, social, technological, environmental, and legal) framework in combination with a SWOT (strengths, weaknesses, opportunities, and threats) analysis. The contribution of this study includes a strategic recommendation to promote the sustainable use of biomass briquettes for industrial purposes, particularly in the tea industry, and an understanding of the use of a combined SWOT/PESTEL analysis in the bioenergy sector.

The article is organized into five sections. Section 1 provides the theoretical and contextual background. Section 2 presents the scope and methods of the study. Section 3 presents the results of the substitution potential and the SWOT/PESTEL analysis. Section 4 discusses the main drivers and barriers to the substitution. Lastly, Section 5 defines the conclusions of the study and provides recommendations for future research.

## 2. Materials and Methods

The design of the study is illustrated in Figure 1. The first part is to analyze the substitution potential, and the second part is to identify the factors influencing the substitution using a combined SWOT/PESTEL analysis based on expert interviews. The results are synthesized to determine the drivers and barriers of the substitution.

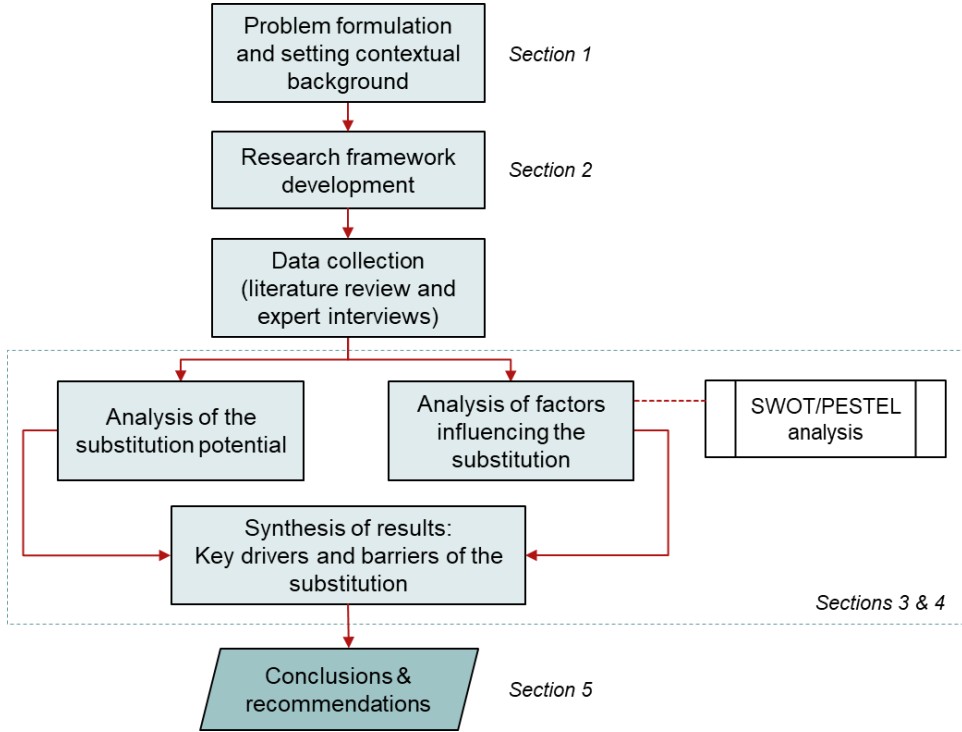

**Figure 1.** Research framework.

The study scope was set within the value chain of firewood and biomass briquette production (Figure 2). This chart was used as an orientation to determine the internal and external factors of the substitution.

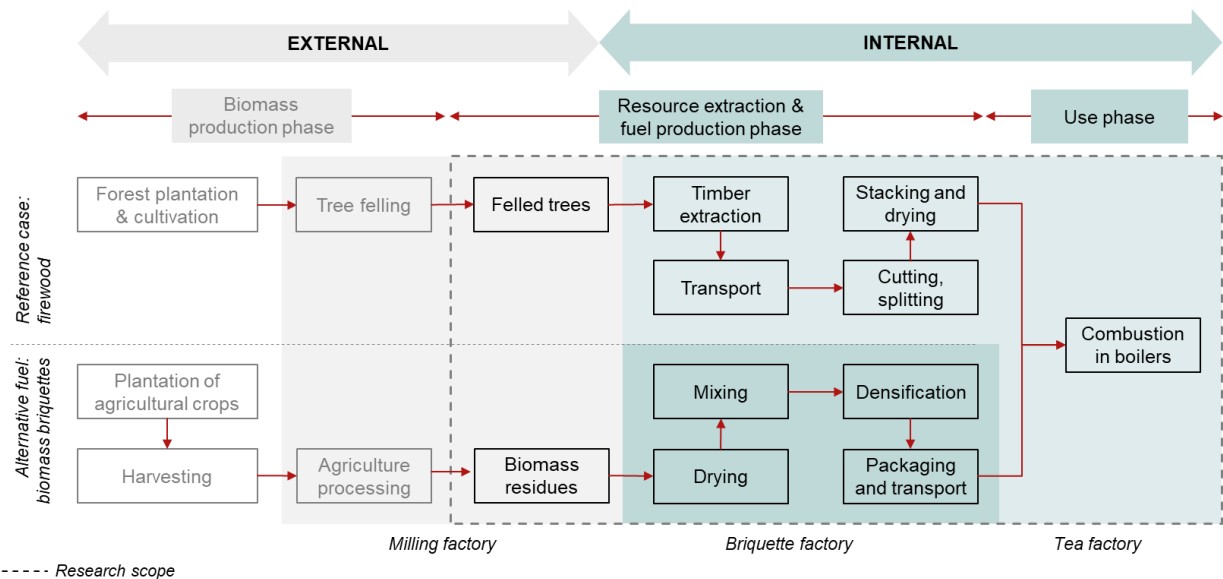

**Figure 2.** External and internal scope in the study.

*2.1. Study Area*

The tea industry in Kenya is operated by the tea estates (involving large plantations) and the smallholder farmers organized by the Kenya Tea Development Agency (KTDA). KTDA is a private holding company, managing an overall 70 factories on behalf of more than 635,000 farmers who have an ownership stake in the factories [16]. Tea is grown in 19 counties located within the West and East of Rift in Kenya, where KTDA operates plantations and factories in 16 counties [16]. Over 60% of the country's tea production comes from KTDA [17,18]. This study, therefore, focuses on the tea factories managed by KTDA.

The study was conducted at the country level. A map (Figure 3) was created by plotting the location data of biomass briquette factories, tea factories, and sugar factories to provide an overview of the geographical information. Only a limited number of briquette factories are shown on the map as the data are scarce. Information on agricultural land and tree plantations is included in the map. The data used to plot the map are provided in the Supplementary Materials.

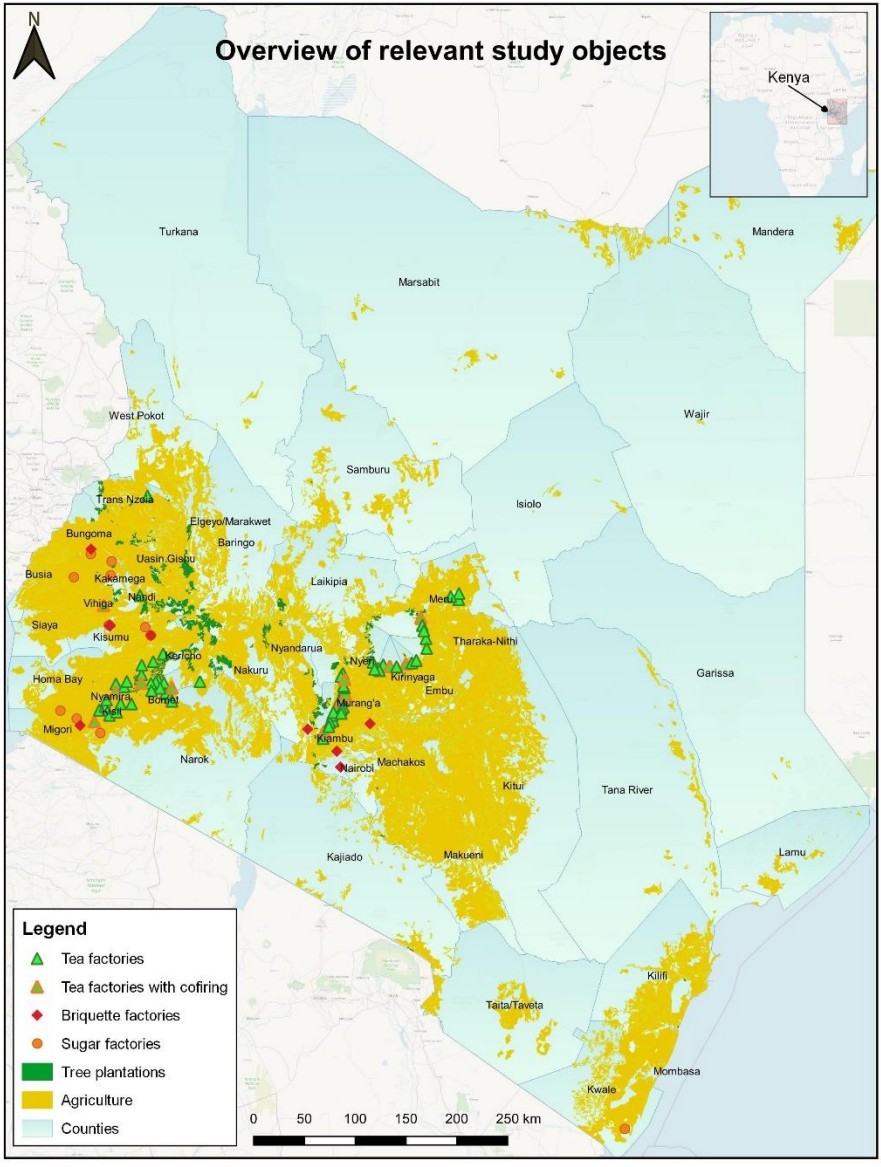

**Figure 3.** Overview of the study objects.

*2.2. Analysis of Substitution Potential*

2.2.1. Biomass Residues Potential

Kenya has potential for biomass residues coming from agricultural products such as sugarcane, maize, potato, cassava, and banana, which are among the country's top ten commodities [19]. The types of biomass residue can be differentiated according to where they are found. Residues such as stalks, stems, straws, and leaves are most likely to be left in the field; thus, they are called field-based residue (F). Meanwhile, residues such as husks, cobs, peelings, nutshells, bagasse, and pulp are typically generated after processing (P), for example, at a milling facility.

Biomass residue potential refers to the number of biomass residues available to be processed into briquettes and the potential energy that can be generated from the residues. The amount of residue was calculated by multiplying the annual crop production quantity by its average RPR (residue-to-product ratio) value (Equation (1)). RPR is a measure of how much residue is generated per mass unit of product [20]. RPR values of a crop might be influenced by its geographic location, crop yield, weather conditions, and moisture content at the time of sampling [21]. The theoretical energy potential refers to the analytical calculations of energy to be generated from the biomass residues based on their characteristics. It was estimated by multiplying the residue quantity by its calorific value (CV) (Equation (2)). Obtaining RPR values that are specific to Kenya is difficult without actually collecting primary data from the field and processing facilities. We adopted the RPR values and CVs from previous studies in Kenya and other countries, as summarized in Tables A1 and A2 of Appendix A.

$$\text{Residues generated (kg)} = \text{Crop production (kg)} \times \text{RPR,} \qquad (1)$$

$$\text{Technical energy potential (MJ)} = \text{Residue quantity (kg)} \times \text{CV (MJ/kg),} \qquad (2)$$

This study considered three aspects in selecting the biomass residues to be further analyzed, namely (1) the abundance of quantity based on the latest data of produced crops, (2) the results of the previous studies [5,20,22], and (3) the consultations with briquette companies in Kenya. The four most promising crops selected for analysis were sugarcane, maize, potato, and banana. Additionally, coffee and rice were included in the analysis despite their relatively small production. This is because several briquette companies are using coffee and rice husks, either as the main raw material or as fillers. Sawdust (i.e., residue from sawmills) is also commonly used in the briquette industry for its high calorific value and is therefore included in the analysis.

2.2.2. Energy Demand in Tea Factories

Energy demand in this study refers to the thermal energy required for the withering and drying processes, which comes from burning firewood in the boilers. The amount of firewood consumed in the tea factories varies according to their production capacity. This study assumed an average annual firewood consumption per factory to be 16,000 m$^3$ per year [23] and the total consumption by all KTDA tea factories to be 900,000 m$^3$ [24]. Since there is no certification scheme in place to measure the moisture content, the density of firewood was assumed. Based on the range of 393–550 kg/m$^3$ [7,24,25], we assumed the density of 460 kg/m$^3$, which results in an equivalent of 414,000 tons of dry firewood demand. This can generate up to 5,796,000 GJ of heat energy, assuming a CV of 14 MJ/kg. The initiative to replace 20–30% firewood with briquettes means that the energy amount required from briquettes would range between 1,159,200 and 1,738,800 GJ. Table A3 of Appendix A provides a list of assumptions used for the calculation.

2.2.3. Briquette Production

A study in Kenya [7] suggested that the level of briquette supply and cost factors may make it unattractive for tea factories to switch from firewood to biomass briquettes.

Therefore, this study examined the briquette production and the costs of purchasing briquettes in comparison to firewood.

ISO 16559:2014 standard defines biomass briquette as "densified biofuel made with or without additives in pre-determined geometric form with at least two dimensions (length, width, height) of more than 25 mm, produced by compressing biomass." Briquetting is the process of compacting residues into a solid product to increase the energy density of the materials, where higher densification leads to less moisture content and higher calorific value compared to its raw materials [26–28].

There are two types of briquettes, i.e., carbonized and non-carbonized. Carbonized briquettes are made from biomass that has undergone pyrolysis, and the feedstock is mixed with a binding agent before being pressed to form briquettes [15]. This type of briquette is preferred for cooking in households as it can light quickly. Non-carbonized briquettes are processed directly from biomass sources through various casting and pressing processes (i.e., densification) and are mainly for industrial use [29]. Non-carbonized briquettes are more favorable for industrial purposes because they can burn for a longer duration.

### 2.2.4. Energy Cost Estimation

This study examined the production capacity and product prices, excluding tax and other costs, based on the data from the literature [7,10] and interviews with briquette producers. The exclusion of other costs was due to the variance of delivery costs and the data scarcity. A sensitivity analysis of the costs of adopting briquettes at different substitution rates in the tea factories and different CVs was conducted. CV is a sensitive parameter to the costs; the higher the CV of briquettes, the more competitive briquettes are to replace firewood as there will be more energy that can be delivered. The sensitivity analysis was carried out to show the importance of having a more standardized CV for briquette products so that the amount of energy to be delivered by briquettes can be ensured. The CV variation was based on the existing data in the literature [24] and the value reported by the briquette companies interviewed for the study (Table A4).

### 2.3. SWOT/PESTEL Analysis

SWOT analysis is commonly used in strategic planning to evaluate an organization, a project, or a business activity based on the internal factors (strengths and weaknesses) and external factors (opportunities and threats) [30]. It has been applied by researchers in the area of energy planning, including the topic of drivers and barriers to renewable energy development [31] and sustainable transition to renewable energy [32]. The SWOT analysis identified and systematically categorized the results into two groups: internal and external factors. Internal factors are those over which the system has control over it and are divided into favorable and unfavorable ones. The favorable internal factors were considered *strengths*, while the unfavorable internal factors were seen as *weaknesses*. With the same logic, the favorable external factors were grouped under *opportunities*, whereas the unfavorable external factors fell under the *threats* category.

A PESTEL analysis was carried out complementary to the SWOT analysis by initially focusing on the key external factors affecting the substitution. It reviewed the macro environment of the transition from firewood to biomass briquettes. PESTEL analysis reflected the political, economic, social, technological, environmental, and legal factors to evaluate the biomass briquettes' potential to substitute firewood in the tea industry. This approach was used to identify the drivers and barriers to the adoption of alternative energy technologies [33] and to analyze the biofuel energy industry in Europe [34].

The combination of SWOT and PESTEL analyses allows for a more thorough and more accurate analysis of a complex system and its multidimensional interactions with the environment [35]. Even though the initial purpose of PESTEL analysis was to comprehend the macro environment of the substitution, the expert interviews revealed that the PESTEL categories were also reflected within the internal factors. Applying a PESTEL analysis to both internal and external factors has been shown in a case study of reconstructing a water

intake structure [35]. The SWOT/PESTEL analysis in this study followed the algorithm as illustrated in Figure 4.

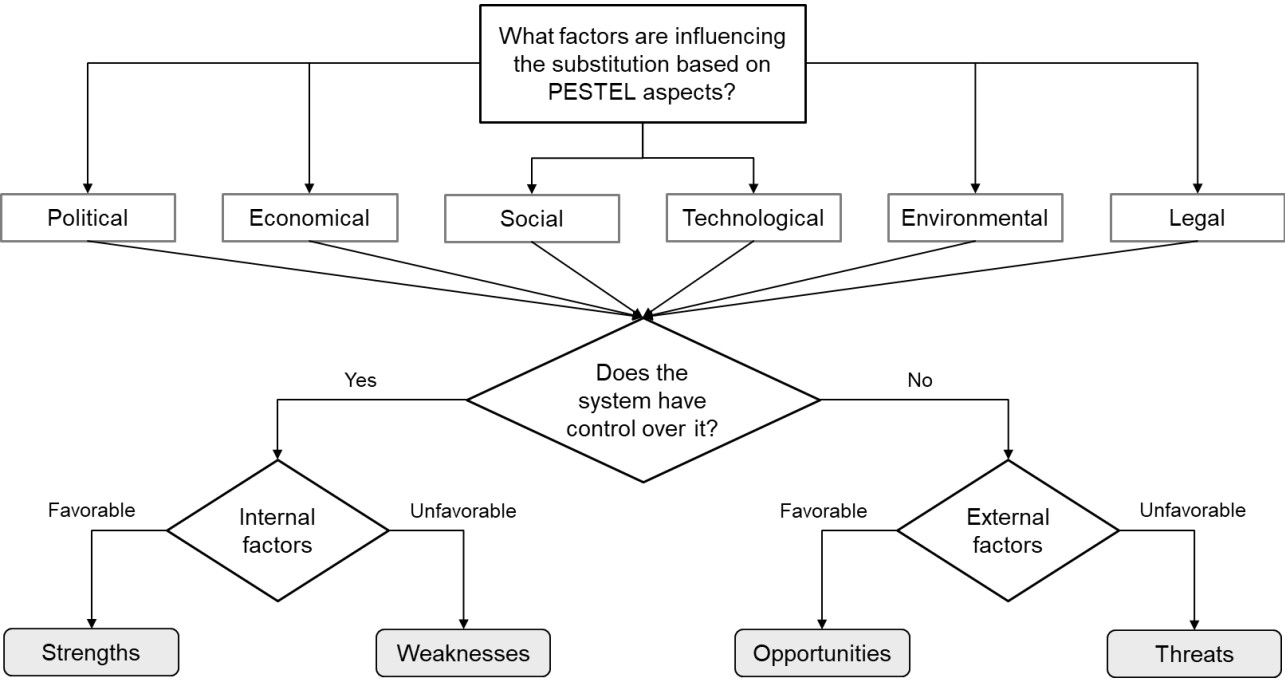

**Figure 4.** Algorithm of the SWOT/PESTEL analysis (Adapted from [32,35]).

The identified factors were categorized according to the political (P), economic (Ec), social (S), technological (T), environmental (En), and legal (L) aspects. Then, they were classified into four SWOT groups based on the logic of internal/external and favorable/unfavorable toward the substitution. In some cases, one factor may indicate multiple SWOT/PESTEL aspects as it is not always easy to allocate one factor to only one category.

Eleven experts were interviewed for this study; five were directors of briquette companies, one was at a managerial position in KTDA, and one was from KEBS (Kenya Bureau of Standards, a government agency dealing with national standards in Kenya), two were from NGOs, one researcher, and one consultant. Nine interviews were recorded, while two were not due to technical issues. There are in total of 365 min of audio recording. All data are confidential and can only be used for academic purposes.

The interviews were evaluated through thematic analysis, a technique for identifying, analyzing, and interpreting patterns of meaning, called *themes*, within qualitative data [36]. This method was chosen for its flexible nature in terms of research questions, sample sizes, data collection methods, and approaches to generating themes [37]. A thematic analysis starts with familiarizing with the data through an iterative process of transcribing the interview recordings into a text form, reading and rereading the transcriptions to comprehend the entirety of the information, and noting down the initial ideas [37]. The raw data were transformed into a standardized format for analysis by identifying and labeling recurrent words and concepts, called *codes*, which are the smallest units of analysis relevant to the research questions [37,38]. From line-by-line coding of the raw data, the generated *codes* were grouped logically to build *themes* as patterns of meaning, known as *categorization* [38]. These steps were also performed iteratively, repeatedly returning to the raw data and adjusting the labeling of *codes* and *themes*. The result of this process is not only a summary of data content but also its interpretation and meaning extracted from the raw data.

### 3. Results

*3.1. Substitution Potential*

3.1.1. Biomass Residues Potential

This study estimated around 18 million tons of the selected biomass residues can be generated every year, resulting in theoretically around 250,000 TJ of energy. Not all of this potential has been utilized as briquettes, mainly because of other uses such as animal feed. The amount of each biomass residue and theoretical energy generation are presented in Table 1. These estimations are largely dependent on the quantity of crop production, the RPR values, and the CV used for calculation.

**Table 1.** Estimated availability of each residue.

| # | Biomass | Residue Type | Type | Production 2019 (Tons/Year) | RPR | CV (MJ/kg) | Potential Residues (Tons/Year) | Theoretical Energy (GJ) |
|---|---------|--------------|------|------------------------------|-----|------------|--------------------------------|-------------------------|
| 1 | Sugarcane | Tops, leaves | F | 4,606,100 | 0.19 | 16.61 | 880,277 | 14,619,198 |
|   | Sugarcane | Bagasse | P | 4,606,100 | 0.38 | 13.72 | 1,750,318 | 24,014,363 |
| 2 | Maize | Stalks | F | 3,897,000 | 1.93 | 12.93 | 7,526,276 | 97,314,750 |
|   | Maize | Husks | P | 3,897,000 | 0.40 | 12.00 | 1,558,800 | 18,705,600 |
|   | Maize | Cobs | P | 3,897,000 | 0.35 | 15.41 | 1,378,889 | 21,248,672 |
| 3 | Potatoes | Stems, leaves | F | 2,000,000 | 0.61 | 16.00 | 1,213,333 | 19,413,333 |
| 4 | Bananas | Leaves, pseudo stems | F | 1,715,770 | 1.68 | 15.33 | 2,873,915 | 44,057,113 |
|   | Bananas | Peelings | P | 1,715,770 | 0.34 | 15.00 | 579,072 | 8,686,086 |
| 5 | Coffee | Husks | P | 44,500 | 0.24 | 14.10 | 10,791 | 152,193 |
| 6 | Rice | Husks | P | 91,845 | 0.27 | 15.03 | 24,415 | 367,013 |
| 7 | Wood | Sawdust | P | NA | NA | 21.65 | 230,000 | 4,979,500 |
|   | **Total** | | | | | | **18,026,087** | **253,557,821** |

F: Field residue; P: Processing residue; RPR: Residue-to-product ratio; CV: Calorific value. Source: Production: FAOSTAT; RPR and CV: see Supplementary Materials Tables S1 and S2.

3.1.2. Briquette Production

There are around 30 briquette companies in the country producing non-carbonized briquettes with vastly differing production capacities. Depending on the weather and the availability of raw materials, briquette companies can produce about 125 to 1500 tons of briquettes per month, assuming an average production level of 5 tons per day with 300 days of operation in one year [7]. Such variation is due to the different sizes of the factories and that one briquette company may have more than one factory. According to five briquette companies interviewed and the previous study [7], a briquette company can produce 125 to 2000 tons per month (Table 2). Taking the median of the known data, which is 350 kg/month, realistically, this leads to a total of 126,000 tons of briquettes that can be produced annually by 30 briquette companies.

**Table 2.** Briquette production capacity.

| Source | Production Capacity | | | Remarks |
|--------|---------------------|----------|----------|---------|
|        | Tons/Day | Tons/Month | Tons/Year | |
| Company 1 | - | 300 | 3600 | Will expand to 500 tons per month |
| Company 2 | 20 | 400 | 4800 | Considering delays, 20 days per month |
| Company 3 | - | 2000 | 24,000 | Information is provided for monthly capacity |
| Company 4 | - | 1667 | 20,000 | Information is provided for annual capacity |
| Company 5 | - | 250 | 3000 | Will expand to 1000 tons per month |

Source: Interviews with briquette producers.

Briquette companies reported different CVs for their products based on product sample tests initiated by the companies themselves, as indicated in Table A4 of Appendix A. With roughly 126,000 tons of briquette production per year, around 1900–2900 TJ of thermal

energy is available to the commercial and industrial sector annually, depending on the CV assumption. This is still less than the demand from the tea factories, which was estimated at 289,800 tons of briquettes annually, assuming the briquette CV at 20 MJ/kg. This estimate also means that the briquette industry in Kenya has the potential to substitute 43% of the firewood demand in KTDA tea factories. While with a moderate scenario where briquettes can only reach a CV of 14 MJ/kg, the substitution potential may decrease to 30% (Table A5 of Appendix A shows the sensitivity of substitution rate depending on CV). However, production capacity alone cannot guarantee that a briquette factory can meet its maximum rate. Moreover, provided the tea and briquette factories' distribution in Kenya (see Figure 3), briquettes are not always feasible for every tea factory because transportation costs would be prohibitive.

### 3.1.3. Energy Cost Estimation

Only two briquette companies shared their product price information, as presented in Table 3, while the remaining data were collected as secondary data. According to Company 5, the briquette price is negotiated, depending on the production costs, mode and costs of transportation (fuel and delivery personnel), and other factors. The CV of briquettes was assumed at 20 MJ/kg. The firewood price was based on the latest data after the moratorium, i.e., $22, whereas the briquette price was the ex-factory price reported by Company 5, i.e., $110, excluding delivery cost. Value-added tax (VAT) was not considered in the calculation, nor other taxes that may be imposed on firewood and briquettes.

**Table 3.** Purchasing price of briquette and firewood.

| Source | Remarks (All Excl. VAT) | | US$ |
|---|---|---|---|
| Company 1 | Delivered price | Per ton | 139 |
| Company 1 | Delivered price | Per ton | 157 |
| Company 5 | Ex-factory price | Per ton | 110 |
| Company 5 | Delivered price | Per ton | 130 |
| Company 5 | Delivered price | Per ton | 170 |
| [7] | Average | Per ton | 90 |
| [7] | Average | Per ton | 39 |
| [10] | Before moratorium | Per $m^3$ | 17 |
| [10] | After moratorium | Per $m^3$ | 22 |
| [24] | - | Per $m^3$ | 19 |

Source: Interviews with briquette companies [7,10,24].

The thermal energy costs of using firewood are estimated at nearly $19.4 million per year for KTDA. Hypothetically, if all firewood is to be replaced with biomass briquettes, with CV at 20 MJ/kg, the costs will increase to around $31.9 million annually. However, this scenario is not realistic since the current briquette production in the country is significantly less, and not all tea factories are close to the briquette factories.

KTDA tea factories have different production capacities and, as a result, varying energy demands. The thermal energy costs of a typical tea factory are estimated to be around $0.35 million per year. Replacing all firewood with briquettes will roughly increase the cost to $0.57 million (as elaborated in Table 4).

**Table 4.** Hypothetical thermal energy costs.

|  | Firewood Only | Unit | Briquette Only | Unit |
|---|---|---|---|---|
| Density [25] | 460 | kg/m$^3$ | 1200 | kg/m$^3$ |
| Calorific value [25] | 14 | MJ/kg | 20 | MJ/kg |
| Total firewood consumption | 16,000 | m$^3$ | *N/A* | - |
| Dry firewood equivalent | 7360 | tons | *N/A* | - |
| Energy consumption | 103,040 | GJ | 103,040 | GJ |
| Briquette equivalent | N/A | - | 5152 | tons |
| Price per unit | 22 | \$/m$^3$ | 110 | \$/ton |
| Total cost (for all factories) | 345,185 | \$ | 566,720 | \$ |

Source: [10,25]; interview; own calculation.

The graph in Figure 5 indicates that replacing firewood with briquettes would cost KTDA more, regardless of the substitution rates and calorific values. Assuming briquette CV = 20 MJ/kg and firewood: briquette ratio of 90:10, energy costs may rise by 6.42% compared to using only firewood; and when the ratio is 80:20, energy costs may rise by 12.84%. Briquette price is a key factor; to be competitive with firewood and for KTDA to pay the same amount of energy costs when replacing 20% firewood consumption, the briquette price needs to go down around half of the current price to \$67 per ton.

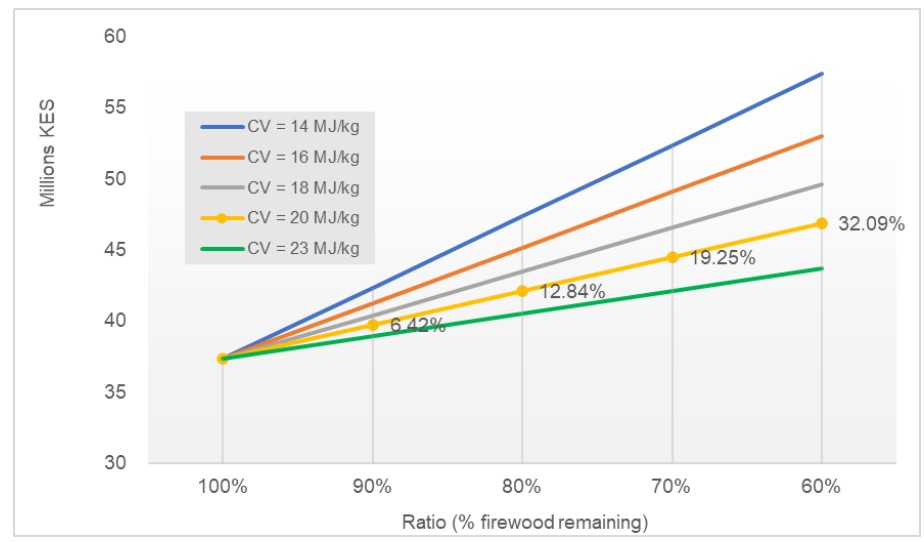

**Figure 5.** Sensitivity analysis on annual thermal energy costs of one factory (own calculation).

### 3.2. Drivers and Barriers of the Substitution

The study identified 27 factors influencing the substitution of firewood with biomass briquettes in the Kenyan tea industry through a SWOT/PESTEL analysis. The strengths and opportunities are considered as drivers, while the weaknesses and threats are assumed to be the barriers to the substitution. Each factor is coded with its PESTEL category, followed by its SWOT category (for example, PO is a political opportunity factor). The summary of all factors is presented in Table 5. The main drivers and barriers are marked in bold font. The identified political, economic, social, technological, environmental, and legal factors are listed in the following sub-sections.

**Table 5.** Matrix of drivers and barriers to substituting firewood with briquettes in the Kenyan tea industry.

| PESTEL Factors | The Transition from Firewood to Biomass Briquettes in the Kenyan Tea Industry | | | |
|---|---|---|---|---|
| | Drivers | | Barriers | |
| | Strengths (S) | Opportunities (O) | Weaknesses (W) | Threats (T) |
| **Political (P)** | — | (1) Policy on forestry<br>- Moratorium on the harvesting of public and commercial forests<br>- Policies to ensure sufficient forest cover of 10%<br>(2) Policy promoting more use of renewable energy i.e., biomass<br>- Proposed financial incentives for both briquette factories and tea factories on procuring equipment<br>**(3) External push and support from development organizations** | - Lack of appropriate financial incentives for businesses<br>- Lack of enforcement on forest harvesting moratorium | **(4) Fiscal policy**<br>- High VAT on briquettes<br>- No VAT on firewood, zero-rated |
| **Economic (Ec)** | (5) High energy demand from the tea industry | (6) The tea's prospective as a prime export commodity<br>- Aspiration to pursue ethical sourcing | (7) Firewood dominance<br>**(8) Lack of stable supply of briquettes**<br>**(9) Cost competitiveness: briquette price is higher than firewood**<br>- Costs associated with the production<br>- Price of raw materials and equipment as compared to unprocessed wood<br>(10) Investment and operational costs<br>- Importation of good briquetting technology requires high investment<br>- Storage and handling | (11) COVID-19 pandemic threatening the business<br>- Depletion of resources in the long-term<br>- Distribution can be problematic (from feedstock location to briquette factories, and then to the tea factories) |

**Table 5.** *Cont.*

| PESTEL Factors | The Transition from Firewood to Biomass Briquettes in the Kenyan Tea Industry | | | |
| | Drivers | | Barriers | |
| | Strengths (S) | Opportunities (O) | Weaknesses (W) | Threats (T) |
| --- | --- | --- | --- | --- |
| **Social (S)** | (12) Ongoing capacity development measures for management, technician, and (boiler) operator level<br>(13) Internal institutional change within KTDA tea companies to look for fuel alternatives | (14) Potential job creation for skilled workers<br>(15) Improved health and safety in using biomass briquettes | (16) Lack of awareness and knowledge regarding biomass briquettes<br>- Changing historic norms from using firewood is happening, but slowly | - Consumer bias, in this case, the KTDA in favor of traditional fuels<br>- Risk of job and/or income loss for firewood suppliers |
| **Technological (T)** | (17) Physical properties of briquettes<br>(18) Staged cofiring<br>- Briquetting is not a new technology in the country<br>- Spare parts and mechanics/technicians are locally available | — | (19) Technical problems with incorporating briquettes in boilers<br>(20) Incompatibility with boiler technology<br>- Different quality of briquettes in the market<br>(21) Deficiency in briquetting technology<br>- Inefficient sun-drying method<br>- Underperforming machines | — |
| **Environmental (En)** | — | **(22) Scarcity of firewood**<br>**(23) Coordinated effort in forest conservation**<br>**(24) Resource availability; biomass potential is enormous** | — | — |
| **Legal (L)** | — | (25) Technical standards on biomass briquettes adopted by KEBS from ISO<br>- Opportunity to apply sustainability standards on tea products regarding the use of sustainable energy | **(26) Lack of awareness of the existing technical standards on biomass briquettes**<br>- Lack of compliance and enforcement of the technical standards | (27) Production processes are not a patentable technology |

### 3.2.1. Political Factors

Political factors refer to government policies and the implementation of policies, which affect the firewood substitution efforts and biomass briquette use in the tea sector.

1. Policy on forestry (PO). Forest conservation has been high on the political agenda in Kenya. One of the actions taken was the moratorium on forest harvesting. Due to the moratorium, firewood has become scarce, resulting in higher prices. As a consequence, the tea factories' interest and urgency to invest in alternative energy sources have grown.

2. Policy on renewable energy (PO). The Parliament of Kenya enacted the Energy Act 2019 to consolidate the laws relating to energy, which includes the promotion of renewable energy. The Act also mandates the promotion of the development and use of renewable energy and providing an enabling framework for efficient and sustainable production, distribution, and marketing of renewable energy, including biomass. Subsequently, the Kenya Bioenergy Strategy 2020–2027 was launched in 2020 to ensure that the use of bioenergy resources is optimized. The Strategy provides a road map for the country to manage and sustainably harness its bioenergy resources [29].

3. External push and support (PO). There has been an external demand from NGOs for tea companies to operate more sustainably. The Rainforest Alliance is one of the key actors in supporting biomass briquette use in tea factories. In 2017, they set up a project that aimed at reducing firewood consumption in tea processing by 30% by using cofiring firewood and biomass briquettes in the boilers with an 80:20 ratio [39]. They also worked with several briquette companies in support of upgrading or upscaling their production as well as to guarantee a captive market, i.e., the tea factories.

4. Fiscal policy (PT). There are three VAT rates in Kenya: 0%, 8%, and 16% [40]. Briquette falls into the general rate for goods and services, i.e., 16%. Imposing VAT on briquettes has been criticized by the sector, considering briquettes are a waste-based commodity. VAT makes briquettes less competitive than firewood. Furthermore, there is no specific financial-related policy to attract tea factories to switch from firewood to biomass briquettes, such as a tax exemption that is applied in the solar and wind energy sector.

### 3.2.2. Economic Factors

Economic factors are those aspects that affect the economic or financial performance of substituting firewood with biomass briquettes in terms of supply and demand, prices, and costs.

5. Energy demand (EcS). Tea processing needs thermal energy for drying and withering the tea, which is currently coming from burning firewood in boilers. Moreover, more tea leaves are expected to be produced in the next years, which means more tea factories are envisaged. Not only the demand from tea factories but also from other large-scale firewood users can be targeted by the briquette companies to expand their business. The challenge is to figure out how much of this demand can be fulfilled by the non-carbonized briquette industry.

6. Tea as a prime commodity (EcO). As a valuable export commodity, there is a lot of interest in making tea a top priority for energy transition efforts. Tea needs to be excellent in quality to be competitive. The global market is increasingly concerned with a product's sustainability, including tea [6,41]. Briquette companies urge RA certification to consider energy use in production as a criterion in the standard. They have a competitive advantage as briquettes are made from waste materials that would otherwise be discarded.

7. Firewood dominance (EcW). KTDA continues to invest in wood fuel plantations to safeguard its energy security. The company emphasizes its environmental sustainability strategy by setting a requirement for its factories to plant one hectare of wood for every four hectares of tea [42].

8. Briquette supply (EcW). Briquette companies acknowledged the challenges in supplying the tea factories with a sufficient quantity of briquettes continuously. The insufficient supply is caused, among other things, by the limited availability of raw materials and drying capacity. Some raw materials have irregular supply. The drying capacity, which still mainly relies on sun-drying, has also been a bottleneck. It is therefore beneficial to implement the cofiring strategy in stages to allow the briquette factories to build up their capacity by improving production facilities and processes.

9. Cost competitiveness (EcW). The cost competitiveness between firewood and briquettes is one of the key factors that hinder the tea industry from moving away from firewood. Briquettes are generally more expensive, depending on the materials and delivery costs. In order to be competitive with firewood, briquettes should have a higher CV and, at the same time, lower price.

10. Investment and operational costs (EcW). From the briquette company's viewpoint, capital investment to procure the equipment is one of the highest costs. Briquette equipment is not readily available in Kenya; hence businesses must rely on imported machines. Since high-quality briquette technology is costly, achieving a return on investment (ROI) is difficult. From the tea factory's viewpoint, when fuel substitution requires a change in boilers, that means an additional investment which makes substitution less appealing. As a result, tea factories strive to mix briquettes with firewood at a rate that does not interfere with the existing boiler system. The operational costs in briquette companies include the costs of raw material collection, drying, processing, and distribution. Transportation is challenging, particularly since briquette factories, which typically are closer to the source of materials, are located far from the tea factories. Collecting sawdust in a large amount is often difficult as well due to the dispersed locations of sawmills, resulting in high transport costs.

11. COVID-19 pandemic (EcT). The pandemic that hit the global population in 2020 affected the biomass briquette sector too. However, among the experts interviewed in this study, only one expressed their concern regarding the pandemic. This is particularly relevant for those briquette producers supplying schools, as they were closed when the lockdown was in place.

3.2.3. Social Factors

Social factors comprise the components that affect the social environment, such as the level of awareness, public acceptance, health benefits, and socio-cultural trends.

12. Capacity development measures (SS). Capacity development is a key strategy to tackle the asymmetry of information and awareness. There are ongoing training activities implemented by the Rainforest Alliance and KTDA on various levels, from the management to the operators. This is a part of sensitization to raise awareness about the cofiring initiative [39]. The goal is to improve the participants' comprehension of the benefits of switching to alternative fuels for production. The boiler operators learned about the technicalities of using briquettes, including how to handle them, how to cofire them with firewood, and what maintenance tasks are needed.

13. Internal institutional change in KTDA (13-SS). KTDA has been proactively preparing its factories to operate more efficiently in terms of energy use. One pilot was implemented in the Makomboki factory in 2015, where the factory installed a briquetting facility to produce briquettes made from a mix of sawdust and macadamia nutshells [43]. Even though KTDA's management understands the urgency to diversify their fuel and move away from the depleting firewood, it is a challenge to inculcate the tea factory staff at all levels with the same understanding.

14. Potential job creation (14-SO). Substituting firewood with biomass briquettes leads to a socio-economic change for the community. On the one hand, briquette production will offer new job opportunities. The switch to briquettes will require a skilled workforce to work in the production process, transportation, and other supporting roles such as administration. On the other hand, switching to briquettes may cause some firewood

suppliers to lose income. Naturally, there is resistance from the firewood suppliers. If there is a substantial increase in firewood replacement with briquettes, the interference in terms of loss of job and income might be bigger.

15. Improved health and safety (SO). Wood combustion emits pollutants of great concern, such as particulate matter, carbon monoxide (CO), and polycyclic aromatic hydrocarbons (PAHs), which pose risks of respiratory illnesses [44]. Biomass briquettes emit less smoke in comparison to firewood [45]. As a result, using briquettes is likely to reduce the risk for those who work near them. Briquettes are also handled differently than firewood, which does not require hard physical work such as chopping down trees, debarking, and splitting the wood. However, there are health and safety risks to the briquette factory workers, such as high concentrations of bagasse dust [7].

16. Lacking awareness and knowledge (SW). Although briquette technology is not new in Kenya, it is known for domestic cooking purposes but is not yet widely recognized as an industrial energy solution. Non-carbonized briquettes only cater to a small niche in industrial processes, whereas the majority still use diesel oil and firewood as fuel. There is an asymmetry of information about the technical and environmental benefits of briquette as well as how to operate it in boilers. More showcases of briquette utilization in the tea industry will help raise the awareness and knowledge level of all actors in the sector.

3.2.4. Technological Factors

Technological factors refer to technical characteristics, the current technology use, and possible innovations in technology that can affect the preference to use biomass briquettes.

17. Physical properties of briquettes (TS). Briquettes have favorable technical properties and do not have much variation in technical parameters. Briquettes are dry, with low moisture content compared to the green wood, which can have around 45% moisture content and requires a months-long seasoning process to dry. Seasoning means storing and stacking firewood under shade to remove its moisture. To obtain a moisture content of $\leq$20%, KTDA recommends a minimum of six months of seasoning [25].

18. Staged cofiring (TS). Cofiring technique is well-known and preferred as a strategy to adopt the utilization of biomass briquettes for tea processing. KTDA has taken up cofiring practices since 2017 in collaboration with the Rainforest Alliance. A total of 18 tea factories are currently attempting to cofire. Briquette companies interviewed in this study suggested the government make it mandatory for tea factories to cofire, based on the rationale that firewood resource is depleting. Through staged cofiring, biomass briquette use will increase gradually and will allow briquette companies to slowly improve their production capacity.

19. Technical problems (TW). Tea factories and briquette companies recognize some technical problems that arise in using biomass briquettes. Firing briquettes, especially the ones made of sugarcane bagasse, have caused the formation of clinkers that may block the aeration in the boiler. Clinker is the solid, stony residue from burning briquettes in the furnace. To manage the risk of clogging due to clinkers, boiler operators have the extra work of poking the briquettes during operation to circulate the air better as well as removing the formed clinkers in the boiler. Such additional maintenance tasks and time lead to more downtime.

20. Boiler incompatibility (TW). When looking back historically, the majority of KTDA tea factories (at least 40 out of 70) were commissioned in the 1950s until the 1980s, and only six factories were built in 2010 or later [46]. The older tea factories used dual-purpose boilers, which were designed for furnace oil and firewood. KTDA is actively aiming for more efficient processes through better insulation and operation, but not necessarily through retrofitting their boilers. Therefore, KTDA's goal is to replace a maximum of 30% firewood through cofiring as it will not require retrofitting.

21. Briquetting technology deficiency (TW). Some concerns related to the technology, among others, are that the machines do not perform as specified, as well as downtime

and breakdown incidents due to poor handling by the machine operator. As part of the production sequence, drying is also crucial. Sun-drying is a common method applied, although it is not reliable, especially in the rainy season. Several briquette companies are pursuing mechanical drying technology when they are about to expand their production capacity.

### 3.2.5. Environmental Factors

Environmental factors refer to the aspects that are determined by the surrounding environment or affect the natural environment.

22. Scarcity of firewood (EnO). The number of firewood resources is dwindling. A report compiled by the Kenya Forestry Research Institute (KEFRI) suggested that the moratorium economically hurt the tea industry. KTDA faced higher production costs as the firewood prices rose, resulting in lower profits paid to the farmers [10]. One tea factory needs around 600 acres of land that is suitable for eucalyptus, and it is challenging to find land that is relatively close to the tea factory area. Vicinity is important because transport costs and pollution related to the longer transport will be problematic if the own forest is too far from the facility.

23. Forest conservation (EnO). This factor is closely linked to the country's efforts to conserve the forests and regain tree cover. The forest sector is as vital as the tea sector, and both are interdependent. Climate change may influence the rainfall pattern, which is crucial for growing tea. Forests are needed to maintain the rainfall, which is crucial for tea plantations, whereas good tree cover will provide the right environment for tea to grow. Substituting firewood with briquettes has the symbiotic effect of protecting trees. Using more biomass briquettes as a replacement for firewood can be a solution to reduce the deforestation rate.

24. Resource availability (EnO). Kenya has abundant agricultural residues suitable for briquettes. However, quantity is only one influencing factor. The location, accessibility, transportation, and current uses also need to be considered. Sugarcane bagasse is the most popular option as it is readily available in large quantities. However, it is mainly found in the western part of the country, known as the "sugar belt," while tea plantations and factories are located in the eastern part. This means that transportation can be costly. Maize husks, cobs, and stalks are abundant in quantity, but these residues are used as animal feed and therefore are not widely processed into briquettes. Sawdust is sold as a waste product and is available in milling regions but is relatively difficult to collect in large volumes.

### 3.2.6. Legal Factors

Legal factors refer to laws and regulations which allow businesses to perform well. The findings include the technical standards by KEBS, its compliance and enforcement, and the issue of patents.

25. Technical standards (LO). KEBS is the government agency that has the mandate to promote standardization in the industry as well as to provide training, testing, and certification services. KEBS has adopted the ISO 17225 series titled "Solid biofuels–Fuel specifications and classes", consisting of seven parts. Part of this series determines the fuel quality classes and specifications of graded non-woody briquettes, which include those made of agricultural residues. The standard specifies the diameter, length, mechanical durability, moisture content, ash content, and calorific value of the briquettes. The classification is based on the source of raw materials.

26. Compliance with technical standards (LW). There are different qualities of briquettes coming into the market as the producers are using different raw materials. Some of these briquettes are sub-standard, which could be due to the different moisture content reached during the drying process as well as different briquetting technology used. This discrepancy in quality is particularly unfavorable to the tea factories as it will affect tea production. The Energy Act 2019 stipulates that everyone should

only use energy products that comply with the KEBS or any international standards approved by KEBS. In reality, however, both briquette producers and consumers are not familiar with these standards. Awareness raising and enforcement efforts about the standards are also still lacking.

27. Non-patentable technology (LT). Briquette production is not straightforward as it consists of several steps, where each process requires certain equipment and know-how. Briquette companies went through a period of trial and error to achieve the optimum sequencing of production steps, which depend on the machines and the raw materials used. There is a concern that such a "secret recipe" for using certain technology and production steps is not a patentable object. This can be a threat for the briquette companies to maintain their competitiveness.

## 4. Discussion

### 4.1. Opportunities as the Driving Force

The drivers for the substitution are found mainly as opportunities in the environmental and political aspects. These include the scarcity of firewood and the difficulties faced by KTDA in acquiring nearby land and planting and growing eucalyptus trees. When the forest harvesting moratorium was enforced, briquette demand increased as the tea factories were pushed to diversify their energy sources. The potential resource availability to make briquettes from residues is also an opportunity for substitution. An appealing aspect of using biomass briquettes is that the raw materials are renewable and derived from otherwise discarded residues. Many local entrepreneurs recognized this potential; by transforming bagasse, sawdust, coffee husks, and rice husks into briquettes. Residues from maize, potato, and banana can further be explored for briquette production in the country.

Politically, the drivers could be better supported by consistent government policies to conserve the forest and promote the use of more biomass residues. Some forms of incentives for both the tea factories and briquette companies were proposed by the experts interviewed in this study. For the tea factories, the support can be in the form of tax holidays and accelerated depreciation so that they can invest in boilers to accommodate more biomass briquettes. Similarly, for briquette companies, a tax holiday to acquire high-quality briquetting equipment was proposed. Additionally, the external push and support from development organizations motivated by sustainability concerns can also further drive the substitution. Such organizations have the vision and resources to bring forward initiatives that may not be the main business interests of the tea factories.

### 4.2. Internal Weaknesses as Barriers to Substitution

Different from the drivers, which mostly are external factors, the barriers to substitution are mainly the internal weaknesses of the briquette product itself. The main barrier discovered was the cost competitiveness. Briquettes are typically more expensive than firewood. In the tea sector, firewood is accessible with a zero-rated tax, while briquettes are subject to 16% VAT. It makes it difficult for briquettes to compete as the production costs are higher than firewood. The experts see the need to advocate for necessary changes in current Kenya's bioenergy strategy and policies to improve the cost competitiveness of biomass briquettes as a firewood replacement. This may include the VAT exemption for briquettes (to be made zero-rated) and financial incentives for the importation of production equipment (as applied to solar and wind technologies).

The supply issue is found as a weakness for the substitution. It includes the quantity and continuity of briquette supply, the production process, and handling. Fetching a sufficient amount of certain raw materials, such as sawdust and coffee husks, is problematic. Meanwhile, for abundant feedstock such as bagasse, the drying process remains heavily reliant on sun-drying. It is debatable whether briquettes are easier to handle than firewood. On the one hand, firewood needs more space to season; a large amount of firewood should be stored under a shed in the factory's vicinity for up to six months, whereas briquettes are more compact and require less space for storage. Briquettes, on the other hand, are

made of residues that can disintegrate if the environment is not conducive to their physical properties. The tea-growing regions are typically wet, which is not ideal for briquettes.

The lack of awareness and knowledge is likely to cause a slow adoption process. Improving the awareness and knowledge level both in the tea and briquette sectors supposedly can expedite the transition. This includes the awareness of existing technical standards adopted by KEBS, which should be widely communicated to all stakeholders. Establishing a quality assurance system for biomass briquettes will increase the tea factories' confidence in incorporating briquettes into their operations. More organized product testing and certification can be introduced. Additionally, a continuous effort to develop more skilled human resources is a prerequisite to expanding the staged cofiring strategy.

The experts had different views regarding the readiness and willingness of the tea factories to switch to biomass briquettes. On the one hand, a group of experts considered that there is no urgency to switch as firewood is still available and cheaper than briquettes. Partially replacing firewood means they have to change some processes to accommodate the inclusion of briquettes. This will influence the boiler operation, handling, storage, and, most importantly, the costs. On the other hand, the remaining experts were convinced that the tea factories were ready to cofire and that shifting from firewood was inevitable. A staged cofiring can alleviate the burden of incremental costs associated with cofiring. The increasing number of factories that participate in the cofiring initiative can be seen as an indication that the willingness to transition is more a driver than a barrier.

### 4.3. Limitations

The findings of this article confirm the results of a previous study regarding barriers to using briquettes [13,14], although both studies did not specifically discuss the use of briquettes in the tea industry. Despite the best efforts to ensure the validity and reliability of the findings, some limitations might lead to inaccuracies in the results concluded in this study. This study was highly reliant on secondary data, including the RPR values, that is scarce in Kenya. The cost analysis conducted in this study did not take into account factors other than the product price. Overall, the outcomes of this work should be regarded as preliminary results that must be further refined as more data are incorporated.

### 5. Conclusions

The tea industry in Kenya could be a prime target for a big energy transition away from firewood and towards biomass briquettes, considering the high value of tea as an export commodity and the high energy demand in its production. Trifold objectives to improve the tea production process, utilize renewable energy, as well as to conserve the forests would be a strategic move. On the supply side, briquette production capacity can still increase if more generated residues are optimally utilized. This can be accomplished by deploying a more efficient manufacturing process, such as mechanical drying, or, in the long run, by using residues that are currently underutilized.

This study revealed several factors that positively influence the substitution. The main drivers include firewood scarcity, biomass resource availability, and external push from development organizations to integrate briquettes through cofiring. The study also concluded the main barriers to substitution are the cost competitiveness, insufficient and non-continuous supply of briquettes, as well as the still lacking awareness and knowledge regarding briquette technology.

The combined SWOT/PESTEL framework worked well in identifying the drivers and barriers factors of an energy transition effort. Defining the system boundary is important to determine the internal and external factors. However, sometimes one factor cannot be fully attributed to only one category, and it may relate to the others. Such a correlation was not part of this paper, thus offering future research potential. The use of primary data in terms of RPR, CV, briquette production capacity, and prices would be beneficial for conducting feasibility studies. A field study on RPR values in Kenya will be valuable to obtain a more accurate picture of residues characteristics specific to the country.

This study has identified the main drivers and barriers to firewood substitution with biomass briquettes in a broad sense. Further research efforts can be further pursued on this topic. These include implementing a comprehensive monitoring system in the tea factories with cofiring to allow for empirical assessment of the energy use, a cost-benefit analysis of the substitution, and case studies on briquette factories supplying to tea factories to show the feasibility of the substitution.

**Supplementary Materials:** The following supporting information can be downloaded at: https://www.mdpi.com/article/10.3390/su14095611/s1, Table S1: Firewood demand. Table S2: Tea factory inventory. Table S3: Location of briquette factories. Table S4: Location of sugar factories. Table S5: List of experts contributing to the study. Table S6: Comparison of physical properties between firewood and briquettes. Table S7: Data sources to establish the map.

**Author Contributions:** Conceptualization, A.S., A.B., C.M.-B. and M.M.; Methodology, A.S. and A.B.; Formal analysis, A.S.; Investigation, A.S.; Resources, A.S. and M.M.; Data curation: A.S.; Writing—original draft preparation, A.S.; Writing—review and editing, A.S., A.B., C.M.-B., D.T. and M.M.; Visualization, A.S.; Supervision: A.B., C.M.-B. and D.T. All authors have read and agreed to the published version of the manuscript.

**Funding:** This work has received financial support from the Helmholtz Association of German Research Centres through the PoF IV Program Changing Earth—Sustaining our Future, Topic 5 Landscapes of the Future.

**Institutional Review Board Statement:** Not applicable.

**Informed Consent Statement:** Not applicable.

**Data Availability Statement:** Not applicable.

**Acknowledgments:** The authors would like to thank all key experts for their participation and contribution to the study, and the German Agency for International Cooperation (GIZ) in Kenya for facilitating the interviews.

**Conflicts of Interest:** The authors declare no conflict of interest.

## Abbreviations

AFA: Agriculture and Food Authority; CV: Calorific value; FAO: Food and Agriculture Organization of the United Nations; ISO: International Organization for Standardization; KEBS: Kenya Bureau of Standards; KTDA: Kenya Tea Development Agency; MC: Moisture content; NGO: Non-Governmental Organization; PESTEL: Political Economic Social Technological Environmental Legal; RPR: Residue-to-product ratio; SWOT: Strengths Weaknesses Opportunities Threats; UNEP: United Nations Environment Programme; VAT: Value Added Tax.

## Appendix A

**Table A1.** RPR values. (F: Field residue; P: Processing residue).

| Crops | Residue | Type | [20] | [20] | [22] | [26] | [21] | [47] | [47] | [48] | [49] | [49] | This Study (Average) |
|---|---|---|---|---|---|---|---|---|---|---|---|---|---|
| Sugarcane | Tops, leaves | F | 0.22 | 0.10–0.33 | - | - | 0.18 | 0.30 | - | 0.11 | 0.13 | 0.05–0.3 | 0.19 |
| Sugarcane | Bagasse | P | 0.38 | 0.36–0.40 | 0.30 | - | - | 0.29 | - | 0.18 | 0.25 | 0.1–1.16 | 0.38 |
| Maize | Stalks, stover | F | 2.70 | - | 0.27 | - | 2.32 | 2.00 | 1.0–2.50 | 1.59 | 1.60 | 1.0–4.33 | 1.93 |
| Maize | Husks | P | - | - | - | - | - | 0.60 | - | 0.20 | - | - | 0.40 |
| Maize | Cobs | P | - | - | 0.20 | - | - | 0.27 | - | 0.29 | 0.30 | 0.2–0.86 | 0.35 |
| Potatoes | Stalks | F | - | - | - | - | - | - | - | - | 0.76 | 0.30–0.76 | 0.61 |
| Bananas | Leaves, pseudo-stems | F | - | - | - | - | 0.35 | - | - | - | 3.00 | - | 1.68 |
| Bananas | Peelings | P | - | - | - | 0.25 | - | - | - | - | 0.35 | 0.35–0.40 | 0.34 |
| Coffee | Husks | P | 0.24 | 0.23–0.25 | 0.25 | - | - | - | - | - | - | - | 0.24 |
| Rice | Husks | P | 0.29 | 0.22–0.35 | 0.20 | - | - | 0.28 | - | 0.26 | - | - | 0.27 |
| | Study location | | Kenya | Kenya | Kenya | Thailand | Columbia | Kenya | Kenya | Ghana | India | India | |

**Table A2.** Calorific values (in MJ/kg).

| Crops | Residue | Type | [13] | [20] | [20] | [22] | [28] | [47] | [49] | [50] | This Study (Average) |
|---|---|---|---|---|---|---|---|---|---|---|---|
| Sugarcane | Tops, leaves | F | - | 16.61 | 15.81–17.41 | - | - | 16.60 | - | - | 16.61 |
| Sugarcane | Bagasse | P | - | 12.93 | 7.75–18.10 | 13.00 | 17.54 | 13.00 | - | - | 13.72 |
| Maize | Stalks, stover | F | - | - | - | 12.50 | 13.79 | 12.50 | - | - | 12.93 |
| Maize | Husks | P | - | - | - | - | - | 12.00 | - | - | 12.00 |
| Maize | Cobs | P | - | - | - | 15.50 | 15.23 | 15.50 | - | - | 15.41 |
| Potatoes | Stalks | F | - | - | - | - | - | - | 16.00 | - | 16.00 |
| Bananas | Leaves, pseudo-stems | F | - | - | - | - | - | - | - | 14.09–16.57 | 15.33 |
| Bananas | Peelings | P | - | - | - | - | - | - | - | 15.00 | 15.00 |
| Coffee | Husks | P | - | 14.10 | 12.20–16.00 | 12.38 | 17.56 | 12.38 | - | - | 14.10 |
| Rice | Husks | P | - | - | 13–19.33 | 13.45 | 13.38 | 16.00 | - | - | 15.03 |
| Sawdust | Sawdust | P | 17.5–34.3 | - | - | 16.32 | 18.48 | - | - | - | 21.65 |
| | Study location | | East Africa | Kenya | Kenya | Kenya | India | Kenya | India | Uganda | |

**Table A3.** List of assumptions for the calculation of substitution potential.

| Parameter | Value | Remarks |
|---|---|---|
| Energy conversion | 1 MJ = 239 kcal | |
| Exchange rate | 1 US$ = 108 KES | KES: Kenyan Shilling |
| Firewood | | |
| Bulk density | 460 kg/m$^3$ | Firewood density is ranging from 393 to 550 kg/m$^3$ in the literature. This study used the number targeted by KTDA [25]. |
| Moisture content (MC) | 20% | KTDA targeted the firewood MC of ≤20%. Greenwood (i.e., freshly cut wood) has ≥30% of MC and can be up to 50%. It should be seasoned for at least 6 months to achieve ≤20% MC. |
| Calorific value (CV) | 14 MJ/kg | KTDA assumed CV at this value. This is considered a moderate value. |
| Energy content | 6440 MJ per m$^3$ | KTDA assumed energy content at 6902 MJ per m$^3$ with the density assumption at 493 kg/m$^3$ [24]. This study used a more moderate assumption on the energy content, derived from the density assumption at 460 kg/m$^3$. |
| Annual consumption for KTDA factories | 900,000 m$^3$ | KTDA assumed an annual firewood consumption of 900,000 m$^3$ for all KTDA factories [24]. |
| Annual consumption for all tea factories | Around 900,000 to 1,000,000 tons | Refs. [7,15]. This is equal to around 1,600,000 to 1,820,000 m$^3$ of firewood, depending on the density assumption. |
| Briquette | | |
| Moisture content (MC) | 10% | KEBS standard requires 12–15% of MC. Briquette companies claimed an MC level between 6 and 14% for their final products. Ref. [24] specified ≤10%. |
| Calorific value (CV) | 20 MJ/kg | Depending on the raw materials, briquette companies claimed the CV of their final products is between 15.9–23 MJ/kg. Ref. [24] assumed an equal CV as firewood at 14 MJ/kg, which is considered too low for briquettes. KEBS requirement is ≥14.5 MJ/kg. |

Source: [7,17,24,25], KEBS standards.

**Table A4.** Briquette properties.

| Source | Moisture Content | Ash Content | Calorific Value (kcal/kg) | | Calorific Value (MJ/kg) | | Raw Materials |
|---|---|---|---|---|---|---|---|
| | | | $CV_{min}$ | $CV_{max}$ | $CV_{min}$ | $CV_{max}$ | |
| Company 1 | 6% | 3% | - | 4252 | - | 17.8 | Bagasse |
| Company 2 | 8–10% | 2% | 4500 | 5000 | 18.8 | 20.9 | Sawdust, coffee husks |
| Company 3 | 8–12% | 5–10% | 3800 | 4200 | 15.9 | 17.6 | Bagasse, sawdust, coffee husks |
| Company 4 | 12–14% | 9–10% | - | 5500 | - | 23.0 | Bagasse, rice/coffee husks as fillers |
| KEBS | 12–15% | <3% | - | - | 14.5 | - | Non-woody briquettes |
| [51] | 9–10% | NA | - | - | 17.4 | 17.8 | Bagasse |

Source: Interviews, company websites, KEBS, [51].

**Table A5.** Substitution potential of firewood with biomass briquettes.

| Assumed Briquette Supply (Tons) | Calorific Value (MJ/kg) | Theoretical Energy Potential (GJ) | Substitution Potential |
|---|---|---|---|
| 126,000 | 14 | 1,764,000 | 30% |
| 126,000 | 16 | 2,016,000 | 35% |
| 126,000 | 18 | 2,268,000 | 39% |
| 126,000 | 20 | 2,520,000 | 43% |
| 126,000 | 23 | 2,898,000 | 50% |

Source: Own calculation.

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
