# Peer review of "Drivers and Barriers to Substituting Firewood with Biomass Briquettes in the Kenyan Tea Industry"

_sustainability, doi:10.3390/su14095611_

Round 1
Reviewer 1 Report
This work assessed the availability and viability of firewood substitution and examines the drivers and barriers of the substitution using a combination of SWOT and PESTEL analysis. The results are very interesting, but there is room for improvement in several areas.
- Appropriate data can be added to the Abstract section.
- The logic of the introduction needs to be cleared up.
- The author needs to provide a clearer picture.
- Page 13 appears as a blank page.
- At the end of the article can the author give a prediction or expectation based on the research?
Author Response
Dear Reviewer,
Thank you for your comments and suggestions. We took your advice for improvement in the following areas:
Point 1: Appropriate data can be added to the Abstract section.
The Abstract has been revised. Some additional relevant information, i.e., SWOT/PESTEL factors, was added.
Point 2: The logic of the introduction needs to be cleared up.
The introduction has been improved. It is now developed with the logic that the tea industry contributes significantly to the Kenyan economy but at the same time risks the forest cover due to its high firewood consumption.
Point 3: The author needs to provide a clearer picture.
The quality of the map (now Figure 3) has been improved. The old picture is replaced with a higher resolution picture.
Point 4: Page 13 appears as a blank page.
It was an error of saving to pdf file. Page 13-14 (now page 11-12) should contain Table 5 which summarizes the PESTEL factors.
Point 5: At the end of the article can the author give a prediction or expectation based on the research?
The outcomes of this work should be regarded as preliminary results that must be further refined as more data is incorporated. In this sense, as several aspects have been identified we have included in the conclusions sections some suggestions on where further research efforts should be focused on.
Furthermore, considering all inputs from all three reviewers, we have done major revisions, particularly on the following items:
- We did a thorough grammar check throughout the document. The title is now slightly corrected.
- Introduction section has been significantly revised, shortened, and sharpened by adding a few new pieces of literature.
- Material and Methods section has been improved to better describe the methodology.
- Considering the lack of depth of the "techno-economic analysis," the term is now rephrased to "substitution potential" which serves as preliminary analysis.
- Results section has been reworked, and a large amount of the texts were removed. The findings of 27 SWOT/PESTEL factors are now numbered.
- Conclusion section has been made more concise and focused on the main findings.
- Supplementary materials will be added.
Reviewer 2 Report
Dear authors,
Thank you for submitting the manuscript.
In this study, the authors identified the factors motivating and hindering the tea factories to use biomass briquettes rather than firewood with the combination of SWOT and PESTEL analysis. It's an intriguing and important study. I have the following suggestions:
- Please add the elaboration of an Abbreviation when used for the first time in the manuscript. Revise the abstract.
- The introduction is too big. Please concise the text to be more critical with less text. There is a lot of information there, but I find it hard to read.
- The methodology also needs concise revision. How techno-economic analysis is done? It requires the equation and the assumption listed in a concise manner.
- Why is page 13 empty?
- The results section needs revision to make it more concise.
- The whole manuscript needs significant revision to make it more concise. There is a lot of redundant text which can be concise in tables or put in the appendix. Please consider that.
Best wishes!
Author Response
Dear Reviewer,
Thank you for your comments and suggestions. We took your advice for improvement in the following areas:
Point 1: Please add the elaboration of an Abbreviation when used for the first time in the manuscript. Revise the abstract.
The firstly-introduced abbreviations have been elaborated throughout the manuscript. The Abstract has been revised.
Point 2: The introduction is too big. Please concise the text to be more critical with less text. There is a lot of information there, but I find it hard to read.
The authors agree that the introduction needs to be more concise, condensing the most relevant background information. We have shortened it to only one page. We also extensively reworked the logic of the introduction, sharpened the most relevant aspects, and added more content regarding the state of research on alternative energy technologies in the tea sector, so that now the problem identification is clearer to the reader. However, considering the importance to provide a contextual background for the study, we shifted some information to the other sections, while keeping in mind its relevance.
Point 3: The methodology also needs concise revision. How techno-economic analysis is done? It requires the equation and the assumption listed in a concise manner.
- The methodology has now been reworked. Figure 2 (now Figure 1) showing the research design has been revised. The term "techno-economic analysis" is rephrased to "analysis of substitution potential" as we acknowledge that the analysis serves as preliminary results for further studies rather than a comprehensive techno-economic analysis.
- The equations are provided and a list of assumptions is provided in Table A3 of the Appendix. Information about Table A3 has been added to the main text.
Point 4: Why is page 13 empty?
It was an error of saving to pdf file. Page 13-14 (now page 11-12) should contain Table 5 which summarizes the SWOT/PESTEL factors.
Point 5: The results section needs revision to make it more concise.
The authors agree that the results section needs to be more concise. This section has been reworked to emphasize the main findings. The 27 SWOT/PESTEL factors are now presented in numbered items. A large amount of narrative text in the results section has been removed.
Point 6: The whole manuscript needs significant revision to make it more concise. There is a lot of redundant text which can be concise in tables or put in the appendix. Please consider that.
The authors thank you for the advice. The manuscript has now been significantly reworked and shortened. As much as possible, redundant texts are removed, and an appendix section has been implemented to provide some relevant supporting information. Table 5 summarized the 27 SWOT/PESTEL, however, we still see the importance of providing an explanation for each of these factors in the manuscript. This explanation has been largely shortened.
Furthermore, considering all inputs from all three reviewers, we have done major revisions, particularly on the following items:
- We did a thorough grammar check throughout the document. The title is now slightly corrected.
- Introduction section has been significantly revised, shortened, and sharpened by adding a few new pieces of literature.
- Material and Methods section has been improved to better describe the methodology.
- Considering the lack of depth of the "techno-economic analysis," the term is now rephrased to "substitution potential" which serves as preliminary analysis.
- Results section has been reworked, and a large amount of the texts were removed. The findings of 27 SWOT/PESTEL factors are now numbered.
- Conclusion section has been made more concise and focused on the main findings.
- Supplementary materials will be added.
Reviewer 3 Report
Title: Drivers and barriers of substituting firewood with biomass briquettes in the Kenyan tea industry
Manuscript No. Sustainability-1675917
General Comments:
Manuscript Id 1675917 entitled “Drivers and barriers of substituting firewood with biomass briquettes in the Kenyan tea industry” has been thoroughly reviewed and my comments are as below:
- The abstract should be revised. Background should be short and give more precise data related to this work.
- Introduction section lacks recent literatures related to the energy conservation techniques in tea industries.
- Authors are advised to provide cost in USD rather than KES throughout the manuscript.
Specific Comments:
- After going through the below table I wonder if the authors have estimated the energy and efforts required for converting wet biomass having more than 50% moisture in briquettes? I think, most of the data given in Table 3 is already known and the authors need to put more effort in calculating the requirement of energy and then NET CV can be calculated?
- In Figure 5, authors have taken CV of briquettes as 14 – 23 MJ/kg. Justify the basis of this much variation in calorific value.
- Conclusion section is too long. This section should include only main findings of this work.
- Formatting of references should be uniform. Besides there are few good articles published on the Barriers, Health and Environmental aspects, they can be consulted for better understanding of the subject and the barriers mentioned in the study could be modified further:
- Comparative study of thermal degradation kinetics of two woody biomass samples for bio-oil production, Sustainable Energy Technologies and Assessments, 522 (2022) 102158.
- The Effects of Household Air Pollution (HAP) on Lung Function in Children: A systematic review, Int. J. Environ. Res. Public Health 18, 11973.
- Experimental and computational investigation of waste heat recovery from combustion device for household purposes, International Journal of Energy and Environmental Engineering, Vol. 12 (2021) 1-12.
- Technological advancements in jaggery-making processes and emission reduction potential via clean combustion for sustainable jaggery production: An overview, Journal of Environmental Management Vol.301 (2022) 113792.
- The methodology of star rating for improved biomass cookstoves: barrier analysis of adoption and plan for remediation of barriers in India and elsewhere. Biofuels, Vol.10 (2019) pp.131-134.
- The authors have mentioned the cost for wood and cost for briquettes, I wonder if they have thought of the environmental issues at the same time, if not, they need to modify their economic and policy issue after including it. For example, the woody biomass is actually non-renewable biomass (wood = NRB) and it is banned in most of the countries but law is not enforced strictly. Therefore, unless we include carbon tax, legal issues, and renewability and carbon neutrality into account, costing can’t be justified.
Recommendation: Overall the manuscript is good but the issues raised here need to be addressed before taking a final call on its suitability for publication.
Author Response
Dear Reviewer,
Thank you for your constructive comments and suggestions. We took your advice for improvement in the following points:
Point 1: The abstract should be revised. Background should be short and give more precise data related to this work.
- The abstract has been revised.
- The authors agree that the introduction needs to be more concise. We have shortened it to only one page. We also extensively reworked the logic of the introduction, sharpened the most relevant aspects, and added more content regarding the state of research on alternative energy technologies in the tea sector. However, considering the importance to provide a contextual background for the study, we shifted some information to the other sections, while keeping in mind its relevance.
Point 2: Introduction section lacks recent literatures related to the energy conservation techniques in tea industries.
We thank you for your comment. We have added recent literature regarding alternative energy technologies in the tea sector. We perceive this course of literature is more closely relevant to the study background than the literature on energy conservation techniques.
Point 3: Authors are advised to provide cost in USD rather than KES throughout the manuscript.
The costs are now provided in USD for the main text of the manuscript, while the information in the supplementary materials is maintained in both currencies.
Point 4: After going through the below table I wonder if the authors have estimated the energy and efforts required for converting wet biomass having more than 50% moisture in briquettes? I think, most of the data given in Table 3 is already known and the authors need to put more effort in calculating the requirement of energy and then NET CV can be calculated?
Table 3 (now Table 1) provides the estimates of the residue potential of the selected biomass. Crop production numbers are known from FAOSTAT data, while the potential residues and theoretical energy were calculated based on the RPR values and CVs assumption. The CVs used in this study are the average of the values from previous studies (a list of previous studies is provided in the Appendix). Determining the net CV has not been the scope of this study as this will require a different methodology.
Point 5: In Figure 5, authors have taken CV of briquettes as 14 – 23 MJ/kg. Justify the basis of this much variation in calorific value.
Thank you for pointing this out. The CV variation was based on the existing data in the literature and the values reported by the briquette companies interviewed for the study (please refer to Table A4 of the Appendix). A justification is added in the text in Section 2.2.
Point 6: Conclusion section is too long. This section should include only main findings of this work.
'The authors agree that the conclusion section needs to be more concise. This section has been reworked to emphasize only the main findings of this work. This work focuses on the drivers and barriers to the substitution based on the SWOT/PESTEL analysis which synthesized the results from a literature review and expert interviews.
Point 7: Formatting of references should be uniform. Besides there are few good articles published on the Barriers, Health and Environmental aspects, they can be consulted for better understanding of the subject and the barriers mentioned in the study could be modified further: (list of papers)
- The reference list is developed automatically by the Mendeley software, using the IEEE style.
- The authors would like to thank you for the suggestion of literature, however, they mostly focus on household uses. We consulted on an article by Hoppin & Jacobs (2013) to enrich the health aspect. This article discusses the health risk of wood combustion in the non-residential sector.
Point 8: The authors have mentioned the cost for wood and cost for briquettes, I wonder if they have thought of the environmental issues at the same time, if not, they need to modify their economic and policy issue after including it. For example, the woody biomass is actually non-renewable biomass (wood = NRB) and it is banned in most of the countries but law is not enforced strictly. Therefore, unless we include carbon tax, legal issues, and renewability and carbon neutrality into account, costing can’t be justified.
- We thank you for this comment. The authors agree that the cost analysis is rather simplistic as it did not include other costs (such as VAT, cess, other taxes, as well as environmental costs). The cost analysis in this study was meant to compare the costs of procuring firewood and briquettes, which was based only on the “purchasing price,” as it was concluded from the expert interviews that the “purchasing price” can lead to high energy costs, thus hindering the tea factories to switch to briquettes. Acknowledging the importance of more thorough cost analysis, we added an explanation of what the analysis means in Section 3.2. We also added a recommendation for future studies on this topic.
- The narration for political and economic aspects was analyzed using the thematic analysis method. The factors were identified based using the SWOT/PESTEL framework derived from expert interviews. To the best of our efforts, we have included all issues mentioned during the interviews.
Round 2
Reviewer 2 Report
Dear authors, Thank you for the revised manuscript. Best of luck!